# Evaluation of Fluconazole versus Echinocandins for Treatment of Candidemia Caused by Susceptible Common *Candida* Species: A Propensity Score Matching Analysis

**DOI:** 10.3390/jof9050539

**Published:** 2023-05-01

**Authors:** Jong Hun Kim, Jin Woong Suh, Min Ja Kim

**Affiliations:** Division of Infectious Diseases, Department of Internal Medicine, Korea University College of Medicine, Seoul 02841, Republic of Korea

**Keywords:** candidemia, fluconazole, echinocandin, mortality

## Abstract

This study aimed to evaluate the effectiveness of fluconazole and echinocandins in the treatment of candidemia caused by both fluconazole- and echinocandin-susceptible common *Candida* species. A retrospective study which enrolled adult candidemia patients ≥19 years diagnosed at a tertiary care hospital in the Republic of Korea from 2013 to 2018 was conducted. Common *Candida* species were defined as *C. albicans*, *C. tropicalis*, and *C. parapsilosis*. Cases of candidemia were excluded based on the following exclusion criteria: (1) candidemia showed resistance to either fluconazole or echinocandins, or (2) candidemia was caused by other *Candida* species than common *Candida* species. In order to compare the mortality rates between patients who receive fluconazole or echinocandins, the propensity scores on variables of baseline characteristics using the multivariate logistic regression analysis were employed to balance the antifungal treatment groups, and a Kaplan–Meier survival analysis was performed. Fluconazole and echinocandins were used in 40 patients and in 87 patients, respectively. The propensity score matching included 40 patients in each treatment group. After matching, the rates of 60-day mortality after candidemia were 30% in the fluconazole group and 42.5% in the echinocandins group, and a Kaplan–Meier survival analysis showed no significant difference between antifungal treatment groups, *p* = 0.187. A multivariable analysis demonstrated that septic shock was significantly associated with the 60-day mortality, whereas fluconazole antifungal treatment was not associated with an excess 60-day mortality. In conclusion, our study results suggest that fluconazole use in the treatment of candidemia caused by susceptible common *Candida* species may be not associated with increased 60-day mortality compared to echinocandins.

## 1. Introduction

Candidemia and deep-seated infection caused by *Candida* species are classified as invasive candidiasis, which is responsible for significant morbidity and mortality in the healthcare setting [1]. Furthermore, candidemia, the most common manifestation of invasive candidiasis, is one of the leading causes of nosocomial bloodstream infections, and the incidence of nosocomial candidemia has increased over recent decades [2,3]. Currently, the Infectious Diseases Society of America (IDSA) guideline recommends an echinocandin as an initial antifungal treatment against candidemia, whereas fluconazole is recommended as an alternative to an echinocandin in selected patients, including those who do not have critical illness and who are considered unlikely to have candidemia with fluconazole resistance [1]. Echinocandins have several favorable features when compared to fluconazole in the treatment of candidemia, including broad-spectrum fungicidal activity with low resistance, safety profile, and few drug interactions [4]. Despite the current recommendation, fluconazole is still used in the treatment of candidemia [5]. Fluconazole also has favorable characteristics including a safety profile, high bioavailability, and low cost [6,7]. There have been limited data regarding the direct comparison between fluconazole treatment and echinocandin treatment for the treatment of candidemia. Only one study of a randomized controlled trial comparing the efficacy of fluconazole and echinocandins for the treatment of invasive candidiasis including candidemia has been identified [8]. In that study, despite the higher treatment response rate in the anidulafungin group than in the fluconazole group (75.6% vs. 60.2%), there was no significant difference in terms of the rate of death between the anidulafungin group and the fluconazole group (23% vs. 31%, *p* = 0.13). Furthermore, despite small numbers of *Candida* isolates with less susceptibility to fluconazole, such isolates were included in the fluconazole group (5 isolates out of 130 isolates), suggesting the additional difficulty in estimation of fluconazole efficacy in relation to echinocandins for treatment of fluconazole-susceptible candidemia. Moreover, most of the recent *Candida* isolates recovered in candidemia in the Republic of Korea (ROK) were susceptible to fluconazole, particularly among common *Candida* species including *C. albicans*, *C. parapsilosis*, and *C. tropicalis* [9]. This raises the question of whether fluconazole could be an acceptable alternative to echinocandins in the treatment of candidemia caused by fluconazole-susceptible *Candida* species in the real clinical setting. As there have been limited data regarding the matter, this study aimed to evaluate the effectiveness of fluconazole and echinocandins in the treatment of candidemia caused by both fluconazole- and echinocandin-susceptible common *Candida* species among hospitalized adult patients in the ROK.

## 2. Materials and Methods

### 2.1. Study Population

A retrospective study of hospitalized patients from 2013 to 2018 was performed at a tertiary care hospital (Korea University Anam Hospital, Seoul, Republic of Korea). Inclusion criteria were the following: (1) adult patients ≥19 years diagnosed with candidemia, (2) candidemia caused by common *Candida* species defined as one of the three *Candida* species (*C. albicans*, *C. parapsilosis*, and *C. tropicalis*), (3) candidemia with proven susceptibility to both fluconazole and echinocandins, and (4) receipt of systemic antifungal treatment with fluconazole or echinocandins for candidemia treatment. Only patients who met all of the inclusion criteria were included in the study. Demographic data and clinical data including comorbidities, clinical conditions, and outcomes were collected. This study was conducted following approval by the institutional review board (IRB Number 2018AN0440) at the Korea University Anam Hospital. Informed consent was waived due to the retrospective nature of the study. 

### 2.2. Definition

The definition of candidemia was at least one positive growth of *Candida* species from peripheral blood culture obtained from an adult hospitalized patient. Treatment against candidemia with a systemic antifungal agent was at the discretion of treating physicians. Administration of fluconazole was based on the renal function. Patients with normal kidney function (defined as creatinine clearance > 50 mL/min) receiving fluconazole were dosed with 800 mg on day 1, followed by 400 mg daily as the maintenance dose. For patients with reduced kidney function (defined as creatinine clearance ≤ 50 mL/min), 400 mg of fluconazole was administered on day 1, followed by 200 mg daily as the maintenance dose. Identification of *Candida* species and determination of antifungal susceptibility were performed using the BacT/ALERT 3D Microbial Detection System (bioMérieux, Inc., Durham, NC, USA) and the automated Vitek 2 Yeast Biochemical Card (bioMérieux, Inc.). Interpretation of antifungal susceptibility was conducted according to the Clinical and Laboratory Standards Institute (CLSI) breakpoints [1]. Patients’ underlying comorbidities were identified, and the Charlson comorbidity index was used for evaluation. Cardiac disease included chronic heart failure and coronary artery disease. Neurologic disease included cerebrovascular disease and dementia. Malignancy included hematologic malignancy and solid organ malignancy. Renal disease included chronic kidney disease and end-stage kidney disease on dialysis. Liver disease included liver cirrhosis. Pulmonary disease included chronic obstructive pulmonary disease and asthma. Clinical conditions including the presence of septic shock, mechanical ventilation, central venous catheter, renal replacement therapy, receipt of parenteral nutrition, neutropenia, recent surgery in the current hospital admission, previous antibiotic therapy within a month, source of candidemia, and use of systemic antifungal treatment were recorded. Neutropenia was defined as an absolute neutrophil count < 500 cells/mm^3^, and the definition of septic shock was adapted from the third International Consensus Definitions for Sepsis and Septic Shock (Sepsis-3) [10]. The outcome of candidemia was 60-day mortality defined as death within 60 days after the first positive blood culture for candidemia. 

### 2.3. Statistical Analysis

Categorical variables were analyzed by Pearson’s chi-square test or Fisher’s exact test. Continuous variables were analyzed with the Mann-Whitney test. Variables with a *p* value < 0.2 in comparison analysis between two antifungal treatment groups of fluconazole and echinocandins were included in a multiple logistic regression analysis to calculate a propensity score [11], which was used for propensity score matching to control for baseline variables. Covariates in calculating the propensity score included underlying comorbidities (cardiac disease, neurologic disease, malignancy, and renal disease), clinical conditions (mechanical ventilation, renal replacement therapy, and presence of septic shock), and source of candidemia due to central venous catheterization. Patients in the fluconazole treatment group were matched at a ratio of 1:1 with those in the echinocandin treatment group by individual propensity scores. In the matched cohort, the Kaplan–Meier survival analysis was performed to assess relationship between antifungal treatment groups and 60-day mortality. A multivariable logistic regression analysis using variables with a *p* value < 0.2 from comparison analysis and antifungal treatment was conducted to estimate the odds ratios (ORs) and 95% confidence intervals (CIs) for 60-day mortality. A *p* value < 0.05 was considered to be statistically significant. SPSS version 23.0 for Windows (SPSS Inc., Chicago, IL, USA) was employed for statistical analyses.

## 3. Results

### 3.1. Baseline Demographics and Clinical Characteristics

During the study period, 223 adult patients diagnosed with candidemia were identified. After application of the inclusion criteria, a total of 127 patients were enrolled in the study (Figure 1). 

The median age of these 127 patients was 71 years with an interquartile range (IQR) of 61–79 years. There were 70 males (55.1%). The patients were categorized into two groups according to the antifungal treatment (fluconazole and echinocandins). There were 40 patients and 87 patients in the fluconazole group and echinocandins group, respectively. In the echinocandins group, micafungin (38 patients) and anidulafungin (42 patients) were used more frequently than caspofungin (7 patients) for the treatment of candidemia. Demographic data of age and sex were similar between the two groups. Regarding underlying comorbidities, there were more patients with neurologic disease (35.6% vs. 17.5%, *p* = 0.038) and renal disease (34.5% vs. 15.0%, *p* = 0.024) in the echinocandins group than in the fluconazole group. For clinical conditions, mechanical ventilator (43.7% vs. 5.0%, *p* < 0.001), renal replacement therapy (26.4% vs. 5.0%, *p* = 0.005), and presence of septic shock (43.7% vs. 22.5%, *p* = 0.022) were also more frequently noted in the echinocandins group. Of note, for 47 patients with septic shock, 38 patients received echinocandins (38/47, 80.9%). There were no significant differences in terms of the distribution of *Candida* species and duration of antifungal treatment. Regarding the length of hospital stay after candidemia, there was no difference between the fluconazole group and the echinocandins group. However, the rate of outcome of 60-day mortality was higher in the echinocandins group than in the fluconazole group (55.2% vs. 30.0%, *p* = 0.008). These results are shown in Table 1.

### 3.2. Propensity-Matched Analysis

To compensate for the differences in the baseline characteristics, variables with a *p* value < 0.2 in comparison analysis between two antifungal treatment groups of fluconazole and echinocandins were included in a multiple logistic regression analysis to calculate a propensity score, which was used for propensity score matching. As a result, patients in the fluconazole treatment group were matched at a ratio of 1:1 with those in the echinocandins treatment group (Figure 1). After the matching, there were no significant differences in terms of underlying comorbidities and clinical conditions between the two groups of fluconazole and echinocandins. The duration of antifungal treatment was similar between the two groups. There was no significant difference in terms of the length of hospital stay after candidemia between the fluconazole group and the echinocandin group. Regarding the 60-day mortality rate, no significant difference was observed between the fluconazole group and the echinocandins group (30.0% vs. 42.5%, *p* = 0.245). These results are shown in Table 1. The Kaplan–Meier curves with the log-rank test indicated that there was no significant difference in 60-day survival (*p* = 0.187) between the fluconazole group and the echinocandins group (Figure 2).

The propensity-matched patients were categorized into two groups according to the 60-day mortality as the survivor group (51 patients, 63.8%) and the non-survivor group (29 patients, 36.3%). Regarding the distribution of *Candida* species, there was a borderline significant trend of more patients with *C. tropicalis* in the non-survivor group than in the survivor group (48.3% vs. 27.5%, *p* = 0.060). Underlying comorbidities were similar between the two groups. However, among clinical conditions, the presence of septic shock (44.8% vs. 11.8%, *p* = 0.001) and neutropenia (17.2% vs. 2.0%, *p* = 0.022) were significantly more prevalent in the non-survivor group than in the survivor group. There was no significant difference regarding the antifungal treatment between the two groups. The proportion of patients with 60-day mortality who received antifungal treatment of fluconazole and echinocandins was 41.4% and 58.6%, respectively, *p* = 0.245. These results are shown in Table 2.

Variables with a *p* value < 0.2 from comparison analysis and antifungal treatment were included in a multiple logistic regression analysis to estimate the ORs and 95% CIs for 60-day mortality. The presence of septic shock (OR 6.545, 95% CI 1.833–23.369, *p* = 0.004) was significantly associated with the 60-day mortality. For the presence of neutropenia, an association with a borderline significance was found (OR 9.987, 95% CI 0.892–111.841, *p* = 0.062). However, there was no significant association between fluconazole or echinocandin antifungal treatment and 60-day mortality (OR 1.259, 95% CI 0.398–3.980, *p* = 0.695). In addition, specific *Candida* species were not significantly associated with the 60-day mortality (for *C. tropicalis*, OR 1.130, 95% CI 0.314–4.067, *p* = 0.852; for *C. parapsilosis*, OR 0.475, 95% CI 0.091–2.491, *p* = 0.379) (Table 3).

## 4. Discussion

Our study using a propensity score analysis demonstrated that antifungal treatment with fluconazole was not associated with higher 60-day mortality compared to echinocandins against candidemia caused by fluconazole-susceptible common *Candida* species in adult patients. Prior to the propensity score matching, prevalence of neurologic disease, renal disease, mechanical ventilation, presence of septic shock, and renal replacement therapy were more common in the echinocandin group. These results might reflect the clinicians’ preference for echinocandins when the clinical condition of candidemia patients is worsening or is critically ill, as recommended by the current IDSA guidelines [1]. In addition, the rate of 60-day mortality in the echinocandins group was higher than that of the fluconazole group in the unmatched cohort. These findings suggest the excess 60-day mortality in the echinocandins group might be secondary to the differences in the clinical conditions, particularly for critical illness such as presence of septic shock. After the propensity score matching controlled for underlying comorbidities and clinical conditions, there were no significant difference in terms of the 60-day mortality between the fluconazole group and echinocandins group. Furthermore, a multivariable logistic regression analysis revealed that presence of septic shock was associated with the 60-day mortality while there was no significant association between fluconazole or echinocandins antifungal treatment and 60-day mortality. Therefore, results from our study support that critical illness such as presence of septic shock may be a more important prognostic factor associated with the 60-day mortality than the type of antifungal treatment, whether fluconazole or echinocandins, against candidemia caused by susceptible common *Candida* species in adult patients. In line with our results, a previous meta-analysis of randomized trials of invasive candidiasis including candidemia reported no significant difference in all-cause mortality between the groups of echinocandins and fluconazole [12]. Similarly, a prior randomized controlled trial comparing efficacy of fluconazole and anidulafungin for treatment of invasive candidiasis including candidemia also showed no significant difference in terms of mortality between the fluconazole group and the anidulafungin group [8]. Furthermore, recent studies [13,14,15] including candidemia patients with septic shock demonstrated that fluconazole antifungal treatment was not associated with increased mortality when compared to echinocandins for treatment of candidemia, suggesting comparability with respect to the mortality outcome. In contrast, a pooled analysis of individual patient data from randomized trials showed that antifungal treatment with echinocandins in patients with candidemia was associated with decreased mortality [16]. However, selection bias from the heterogeneity from the study design may lead to limited applicability [17]. Another study conducted among critically ill patients with candidemia revealed that the use of echinocandins was a protective factor for 30-day and 90-day mortality when compared to fluconazole [18]. However, 10 patients in the fluconazole group (10/115, 8.7%) received inappropriate antifungal treatment as *Candida* species isolated from these patients were resistant to fluconazole, which might have contributed to the superior efficacy of echinocandins. On the contrary, all antifungal treatments in our study were considered to be appropriate as only *Candida* species with proven susceptibility to both fluconazole and echinocandins were included. Thus, equal distribution of appropriateness of antifungal treatment between fluconazole group and echinocandin group might have led to the outcome of 60-day mortality observed in our study. In summary, extrapolating from our results suggests that fluconazole could be a reasonable alternative to echinocandins in the treatment of candidemia caused by susceptible common *Candida* species.

Previous studies [13,19] reported that the severity of organ dysfunction manifested as the Sequential Organ Failure Assessment (SOFA) score and Acute Physiology or Chronic Health Evaluation II (APACHE II) score was one of the independent predictors of mortality in candidemia patients. Both SOFA score and APACHE II score are widely used for indicating the severity of septic shock [20,21]. In accordance with these results, our study identified the presence of septic shock to be significantly associated with 60-day mortality. Additionally, neutropenia was found to be another potential risk factor associated with the 60-day mortality in our study with a borderline significance. This finding was consistent with a previous study [22], which showed that neutropenia and APACHE score were predictors of mortality in candidemia patients. Therefore, the apparent association between clinical conditions and the 60-day mortality in our study could be explained by postulating that host-related factors may be the most important predictors of the mortality outcome.

Our study has several limitations. This was a single-center retrospective study. Thus, the sampling bias introduced by the small number of patients, confounding effects from unmeasured variables such as receipt of antifungal treatment prior to the episode of candidemia, and selection bias from preferences of treating physicians might have affected our analysis. In addition, we were not able to include other components of candidemia management that might have influenced the mortality outcome such as adequate source control. However, we used constant definitions and employed multivariable logistic regression analysis to overcome the potential bias. In addition, the retrospective nature of our study did not allow randomization of the patients according to the antifungal treatment. Nonetheless, the propensity score matching was used to control the baseline characteristics of the patients by the antifungal treatment groups in an attempt to minimize residual confounding.

## 5. Conclusions

In conclusion, our study showed that fluconazole use in the treatment of candidemia caused by susceptible common *Candida* species may be not associated with increased 60-day mortality compared to echinocandins. These results suggest that fluconazole may be a possible alternative to echinocandins in the treatment of candidemia caused by susceptible common *Candida* species. Prospective studies involving larger numbers of candidemia patients are required to define the appropriate role of fluconazole in the treatment of candidemia.

## Figures and Tables

**Figure 1 jof-09-00539-f001:**
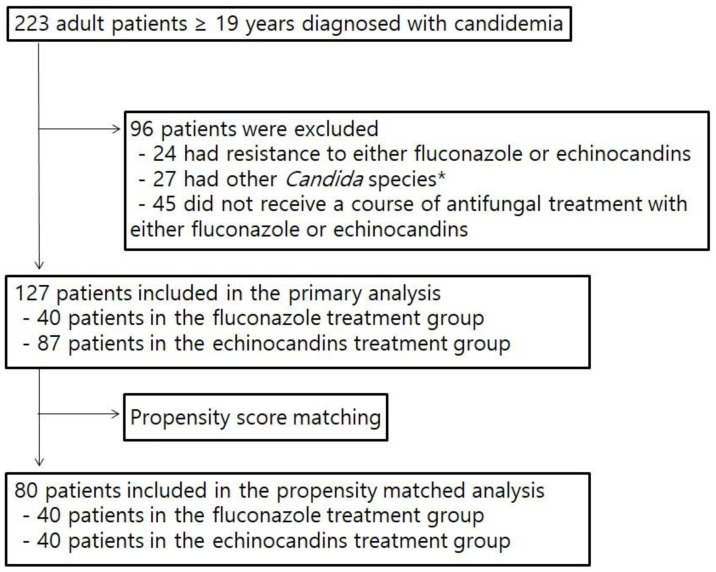
Flow diagram of the study patients included in the study. * Other *Candida* species include *C. glabrata*, *C. haemulonii*, *C. guilliermondii*, *C. utilis*, and *C. lusitaniae*.

**Figure 2 jof-09-00539-f002:**
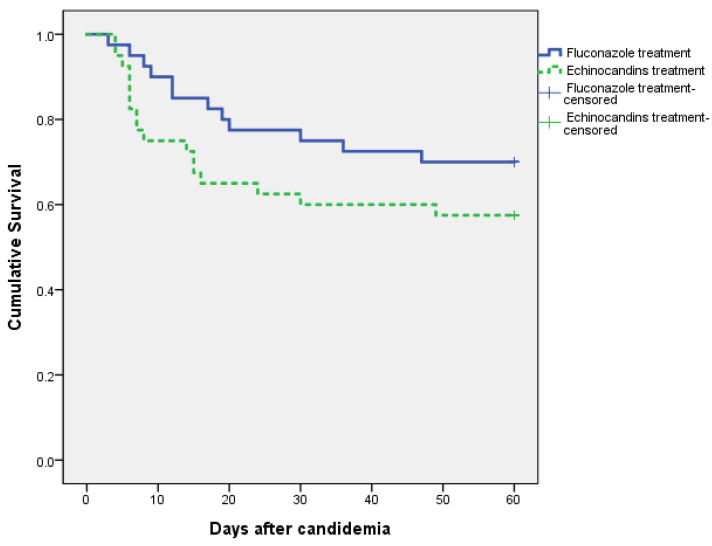
Kaplan–Meier survival curve (60-day survival) after initiation of antifungal therapy with fluconazole or echinocandins.

**Table 1 jof-09-00539-t001:** Baseline and following matching characteristics of candidemia patients by antifungal treatment with fluconazole.

	Whole Cohort			Propensity Score-Matched Cohort		
	Fluconazole Treatment N = 40	Echinocandins Treatment N = 87	*p* Value	Fluconazole Treatment N = 40	Echinocandins Treatment N = 40	*p* Value
Age median (IQR ^1^)	69 (58–81)	72 (63–79)	0.880	69 (58–81)	71 (65–79)	0.844
*Sex*			0.986			0.822
Male, n (%)	22 (55.0)	48 (55.2)		22 (55.0)	23 (57.5)	
Female, n (%)	18 (45.0)	39 (44.8)		18 (45.0)	17 (42.5)	
*Comorbid illness*						
Cardiac disease, n (%)	10 (25.0)	37 (42.5)	0.057	10 (25.0)	15 (37.5)	0.228
Neurologic disease, n (%)	7 (17.5)	31 (35.6)	0.038	7 (17.5)	8 (20.0)	0.775
Malignancy, n (%)	24 (60.0)	40 (46.0)	0.142	24 (60.0)	25 (62.5)	0.818
Renal disease, n (%)	6 (15.0)	30 (34.5)	0.024	6 (15.0)	9 (22.5)	0.390
Liver disease, n (%)	2 (5.0)	12 (13.8)	0.223	2 (5.0)	2 (5.0)	1.000
Pulmonary disease, n (%)	5 (12.5)	12 (13.8)	0.842	5 (12.5)	6 (15.0)	0.745
Diabetes mellitus, n (%)	19 (47.5)	31 (35.6)	0.204	19 (47.5)	17 (42.5)	0.653
Charlson comorbidity index, median (IQR)	2 (1–6)	2 (1–5)	0.381	2 (1–6)	2 (1–7)	0.872
*Candida species*						
*C. albicans*, n (%)	22 (55.0)	44 (50.6)	0.643	22 (55.0)	14 (35.0)	0.072
*C. tropicalis*, n (%)	10 (25.0)	28 (32.2)	0.412	10 (25.0)	18 (45.0)	0.061
*C. parapsilosis*, n (%)	8 (20.0)	15 (17.2)	0.708	8 (20.0)	8 (20.0)	1.000
*Clinical condition*						
Previous *Candida* species colonization, n (%)	0 (0.0)	1 (1.1)	1.000	0 (0.0)	0 (0.0)	NA ^2^
Previous use of antibiotics, n (%)	37 (92.5)	78 (89.7)	0.752	37 (92.5)	33 (82.5)	0.176
Central venous catheterization, n (%)	23 (57.5)	60 (69.0)	0.207	23 (57.5)	25 (62.5)	0.648
Total parenteral nutrition, n (%)	36 (90.0)	84 (96.6)	0.205	36 (90.0)	38 (95.0)	0.675
Use of ventilator, n (%)	2 (5.0)	38 (43.7)	< 0.001	2 (5.0)	4 (10.0)	0.675
Renal replacement therapy, n (%)	2 (5.0)	23 (26.4)	0.005	2 (5.0)	2 (5.0)	1.000
Neutropenia, n (%)	1 (2.5)	7 (8.0)	0.434	1 (2.5)	5 (12.5)	0.201
Presence of septic shock, n (%)	9 (22.5)	38 (43.7)	0.022	9 (22.5)	10 (25.0)	0.793
Recent surgery in the current admission prior to candidemia, n (%)	13 (32.5)	20 (23.0)	0.256	13 (32.5)	11 (27.5)	0.626
*Source of candidemia*			0.159			0.260
Central venous catheter, n (%)	20 (50.0)	55 (63.2)		20 (50.0)	25 (62.5)	
Others ^3^, n (%)	20 (50.0)	32 (36.8)		20 (50.0)	15 (37.5)	
*Antifungal treatment*						
Duration of treatment (days), median (IQR)	14 (7–17)	13 (5–16)	0.216	14 (7–17)	13 (5–16)	0.402
*Treatment Outcome*						
Length of hospital stay (days) after candidemia, median (IQR)	22 (12–35)	20 (9–47)	0.510	22 (12–35)	20 (8–45)	0.570
Mortality day 60 after candidemia, n (%)	12 (30.0)	48 (55.2)	0.008	12 (30.0)	17 (42.5)	0.245

^1^ IQR, interquartile range; ^2^ NA, not available, ^3^ Others, other sources of candidemia including abscess, gastrointestinal tract, urinary tract, and unknown.

**Table 2 jof-09-00539-t002:** Characteristics of propensity-matched candidemia patients stratified to 60-day mortality.

	Survivor N = 51	Non-Survivor N = 29	*p* Value
*Comorbid illness*			
Cardiac disease, n (%)	16 (31.4)	9 (31.0)	0.975
Neurologic disease, n (%)	9 (17.6)	6 (20.7)	0.737
Malignancy, n (%)	30 (58.8)	19 (65.5)	0.555
Renal disease, n (%)	8 (15.7)	7 (24.1)	0.352
Liver disease, n (%)	3 (5.9)	1 (3.4)	1.000
Pulmonary disease, n (%)	5 (9.8)	6 (20.7)	0.193
Diabetes mellitus, n (%)	24 (47.1)	12 (41.4)	0.624
Charlson comorbidity index, median (IQR ^1^)	2 (1–6)	2 (1–7)	0.964
*Clinical condition*			
Previous *Candida* species colonization, n (%)	0 (0.0)	0 (0.0)	NA ^2^
Previous use of antibiotics, n (%)	45 (88.2)	25 (86.2)	1.000
Central venous catheterization, n (%)	25 (49.0)	23 (79.3)	0.008
Total parenteral nutrition, n (%)	46 (90.2)	28 (96.6)	0.409
Use of ventilator, n (%)	3 (5.9)	3 (10.3)	0.662
Renal replacement therapy, n (%)	2 (3.9)	2 (6.9)	0.618
Neutropenia, n (%)	1 (2.0)	5 (17.2)	0.022
Presence of septic shock, n (%)	6 (11.8)	13 (44.8)	0.001
Recent surgery in the current admission prior to candidemia, n (%)	17 (33.3)	7 (24.1)	0.388
*Source of candidemia*			0.084
Central venous catheter, n (%)	25 (49.0)	20 (69.0)	
Others ^3^, n (%)	26 (51.0)	9 (31.0)	
*Candida species*			
*C. albicans*, n (%)	24 (47.1)	12 (41.4)	0.624
*C. tropicalis*, n (%)	14 (27.5)	14 (48.3)	0.060
*C. parapsilosis*, n (%)	13 (25.5)	3 (10.3)	0.104
*Antifungal treatment*			0.245
Fluconazole, n (%)	28 (54.9)	12 (41.4)	
Echinocandins, n (%)	23 (45.1)	17 (58.6)	

^1^ IQR, interquartile range; ^2^ NA, not available, ^3^ Others, other sources of candidemia including abscess, gastrointestinal tract, urinary tract, and unknown.

**Table 3 jof-09-00539-t003:** Factors associated with 60-day mortality.

	Odds Ratio	95% Confidence Interval	*p* Value
*Antifungal treatment*			
Fluconazole (reference)			
Echinocandins	1.305	0.408–4.174	0.653
*Source of candidemia*			
No central venous catheter (reference)			
Central venous catheter	0.804	0.166–3.894	0.786
*Candida species*			
No *C. tropicalis* (reference)			
*C. tropicalis*	1.130	0.314–4.067	0.852
No *C. parapsilosis* (reference)			
*C. parapsilosis*	0.475	0.091–2.491	0.379
*Comorbid illness*			
No pulmonary disease (reference)			
Pulmonary disease	2.533	0.518–12.379	0.251
*Clinical condition*			
No central venous catheterization (reference)			
Central venous catheterization	3.241	0.644–16.312	0.154
No neutropenia (reference)			
Neutropenia	9.987	0.892–111.841	0.062
No presence of septic shock (reference)			
Presence of septic shock	6.545	1.833–23.369	0.004

## Data Availability

The data presented in this study are available on request from the corresponding author.

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
