# Peer review of "Evaluation of Fluconazole versus Echinocandins for Treatment of Candidemia Caused by Susceptible Common *Candida* Species: A Propensity Score Matching Analysis"

_jof, 2023, doi:10.3390/jof9050539_

Round 1

Reviewer 1 Report

The study by Kim J et al.provide valuable data regarding comparison Fluconazole and Ecinocandin in treatment of Candidemia.

Below some comments should be answered by authors to improve the quality of study .

1.Why the authors used just common Candida species specially three of species ?

2.what was the duration of staying in the hospitals in these patients included in this study ?

3.In Table 1 clarify was was the other source of candidemia in the patients ?

4.What percentage of patients  had a history of fluconazole treatment ?

5.How was the dosage of fluconazole for treatment ,please clarify it

5.Discussion needs more relevant studies to support use of fluconazole as a first line therapy and superiority that echinocandin in candedemia .

Author Response

The study by Kim J et al.provide valuable data regarding comparison Fluconazole and Ecinocandin in treatment of Candidemia.

Below some comments should be answered by authors to improve the quality of study .

1.Why the authors used just common Candida species specially three of species ?

-> As stated in the introduction, the most common Candida isolates identified from candidemia in the Republic of Korea were C. albicans, C. parapsilosis, and C. tropicalis, and the most of these Candida isolates were found to be susceptible to fluconazole. However, there has been limited data regarding comparison of antifungal treatment between fluconazole and echinocandins for treatment of candidemia caused by these fluconazole-susceptible common Candida species. That is the reason we decided to conduct our study.     

2.what was the duration of staying in the hospitals in these patients included in this study ?

-> As requested by the reviewer, we performed an additional analysis. In the unmatched cohort, there was no significant difference between the fluconazole group and echinocandins group (median 22 days vs. 20 days, P=0.510). Also, in the matched cohort, no significant difference was found between the fluconazole group and echinocandins group (median 22 days vs. 20 days, P=0.570). We added this information in the results.

“…Regarding the length of hospital stay after candidemia, there was no difference between the fluconazole group and echinocandins group…”

“…There was no significant difference in terms of the length of hospital stay after candidemia between the fluconazole group and echinocandins group…”

3.In Table 1 clarify was was the other source of candidemia in the patients ?

-> Others indicate other sources of candidemia including abscess, gastrointestinal tract, urinary tract, and unknown. We added this information in the footnote of the Table1 and Table2.

4.What percentage of patients had a history of fluconazole treatment ?

-> Unfortunately, we did not collect the history of fluconazole treatment prior to the episode of candidemia. Also, we are unable to collect such information at this time under the circumstances and we recognize this could be one of limitations of our study. Thus, we added this aspect as one of limitations of our study in the discussion.

“…confounding effects from unmeasured variables such as receipt of antifungal treatment prior to the episode of candidemia…”

5.How was the dosage of fluconazole for treatment ,please clarify it

-> The dosage of the fluconazole used in our study for candidemia treatment was 800mg on the first day, then 400mg per day for the patients with normal kidney function (creatinine clearance > 50 mL/min). For the patients with reduced kidney function (creatinine clearance ≤ 50mL/min), 50% of dose used as 400mg on the first day, then 200mg per day. We added this information in the methods.

“…Administration of fluconazole was based on the renal function. Patients with normal kidney function (defined as creatinine clearance > 50 mL/min) receiving fluconazole were dosed with 800mg on day 1, followed by 400mg daily as the maintenance dose. For patients with reduced kidney function (defined as creatinine clearance ≤ 50mL/min), 400mg of fluconazole was administered on day 1, followed by 200mg daily as the maintenance dose…” 

5.Discussion needs more relevant studies to support use of fluconazole as a first line therapy and superiority that echinocandin in candedemia .

-> There has been a limited data regarding direct comparison study between fluconazole and echinocandins. Only one study of RCT comparing the efficacy of fluconazole and echinocandins regarding treatment of invasive candidiasis including candidemia has been identified [N Engl J Med. 2007 Jun 14;356(24):2472-82]. In that study, treatment response rate was higher in the anidulafungin group than in the fluconazole group (75.6% vs. 60.2%). However, the rate of death was 31% in the fluconazole group and 23% in the anidulafungin group without statistical significance (P=0.13). Moreover, only a few retrospective studies have been conducted and we already included these studies in our discussion. However, as recommended by the reviewer, we added an aforementioned RCT study [N Engl J Med. 2007 Jun 14;356(24):2472-82] in the discussion.

“…There has been limited data regarding the direct comparison between fluconazole treatment and echinocandins treatment for treatment of candidemia. Only one study of randomized controlled trial comparing the efficacy of fluconazole and echinocandins for treatment of invasive candidiasis including candidemia has been identified [8]. In that study, despite of the higher treatment response rate in the anidulafungin group than in the fluconazole group (75.6% vs. 60.2%), there was no significant difference in terms of the rate of death between the anidulafungin group and the fluconazole group (23% vs. 31%, P=0.13)…”

“…Similarily, a prior randomized controlled trial comparing efficacy of fluconazole and anidulafungin for treatment of invasive candidiasis including candidemia also showed no significant difference in terms of mortality between fluconazole group and anidulafungin group [8]…”

As stated in the conclusion, our results did not support the superiority of fluconazole in the treatment of candidemia. However, we believe that our results showed that fluconazole may be a possible alternative to echinocandin in the treatment of candidemia caused by susceptible common Candida species, suggesting the possible non-inferiority of fluconazole in the treatment of candidemia caused by susceptible common Candida species.

“…These results suggest that fluconazole may be a possible alternative to echinocandins in the treatment of candidemia caused by susceptible common Candida species. …””

Reviewer 2 Report

The study evaluated the efficacy of fluconazole versus echinocandins for candidemia treatment by score-matching analysis. Despite the potential application of the study for candidemia treatment, there are several concerns as followed:

Major comments:

1.      (Line 68-71) Exclusion criteria is not to be just opposite of the inclusion criteria. The inclusion and exclusion criteria have to be rewritten.

2.      (Line 100-117) As there is more than one way to perform the score-matching analysis, standard references have to be cited for all steps in the statistical analysis.

3.      (Line 165-173) Score matching analysis is specific to a certain research question. Therefore, the recategorization is not acceptable.

Minor comments:

1.      Information on the efficacy of both drugs for candidemia treatment has to be added in the introduction. Specifically, is there any RCT comparing the efficacy of fluconazole and echinocandins?

2.      (Line 86-93) These detail should be moved to the result part.

Author Response

The study evaluated the efficacy of fluconazole versus echinocandins for candidemia treatment by score-matching analysis. Despite the potential application of the study for candidemia treatment, there are several concerns as followed:

Major comments:

  1. (Line 68-71) Exclusion criteria is not to be just opposite of the inclusion criteria. The inclusion and exclusion criteria have to be rewritten.

-> Exclusion criteria may seem to be opposite of the inclusion criteria. However, the purpose of the exclusion criteria in our study was to ensure for all study participants to have all the inclusion criteria. Therefore, we believe that our exclusion was necessary for our study to be conducted in the correct and desired way. Furthermore, the fact that other two reviewers do not have concerns regarding the exclusion criteria supports that the exclusion criteria in the current format may be used in our study.  

  1. (Line 100-117) As there is more than one way to perform the score-matching analysis, standard references have to be cited for all steps in the statistical analysis.

-> As suggested by the reviewer, we added a reference for the calculation of propensity score.

“…to calculate a propensity score [11]…”

References

  1. Austin PC. An Introduction to Propensity Score Methods for Reducing the Effects of Confounding in Observational Studies. Multivariate Behav Res. 2011 May;46(3):399-424.
  2. (Line 165-173) Score matching analysis is specific to a certain research question. Therefore, the recategorization is not acceptable.

-> Score matching was conducted for 1:1 matching comparison between fluconazole treatment group and echinocandins treatment group. As a result of score matching, there was a 1:1 matched cohort of fluconazole treatment group and echinocandins treatment group. Although our primary research question was to have comparison analysis between matched cohort of fluconazole treatment group and echinocandins treatment group, we also wanted to investigate the factors associated with 60-day mortality in the 1:1 matched cohort. In order to proceed, in the 1:1 matched cohort of fluconazole treatment group and echinocandins treatment group, categorization between 60-day survivor and non-survivor was needed for 60-day mortality analysis. Thus, we believe that categorization between 60-day survivor and non-survivor in the 1:1 matched cohort was a necessary step for the 60-day mortality analysis in the 1:1 matched cohort.

Minor comments:

  1. Information on the efficacy of both drugs for candidemia treatment has to be added in the introduction. Specifically, is there any RCT comparing the efficacy of fluconazole and echinocandins?

-> Only one study of RCT comparing the efficacy of fluconazole and echinocandins regarding treatment of invasive candidiasis including candidemia has been identified [N Engl J Med. 2007 Jun 14;356(24):2472-82]. In that study, treatment response rate was higher in the anidulafungin group than in the fluconazole group (75.6% vs. 60.2%). However, the rate of death was 31% in the fluconazole group and 23% in the anidulafungin group without statistical significance (P=0.13). Furthermore, despite of small numbers of Candida isolates with less susceptibility to fluconazole, such isolates were included in the fluconazole group (5 isolates out of 130 isolates), suggesting the additional difficulty in estimation of fluconazole efficacy in relation to echinocandins for treatment of fluconazole-susceptible candidemia. We added this information in the introduction as recommended by the reviewer.

“…There has been limited data regarding the direct comparison between fluconazole treatment and echinocandins treatment for treatment of candidemia. Only one study of randomized controlled trial comparing the efficacy of fluconazole and echinocandins for treatment of invasive candidiasis including candidemia has been identified [8]. In that study, despite of the higher treatment response rate in the anidulafungin group than in the fluconazole group (75.6% vs. 60.2%), there was no significant difference in terms of the rate of death between the anidulafungin group and the fluconazole group (23% vs. 31%, P=0.13). Furthermore, despite of small numbers of Candida isolates with less susceptibility to fluconazole, such isolates were included in the fluconazole group (5 isolates out of 130 isolates), suggesting the additional difficulty in estimation of fluconazole efficacy in relation to echinocandins for treatment of fluconazole-susceptible candidemia…”

  1. (Line 86-93) These detail should be moved to the result part.

-> These details are regarding definition of clinical variables which are needed to be collected. Thus, we believe that these details belong to the components of the definition in the Materials and Methods section.

Reviewer 3 Report

I would like to congratulate the authors on this excellent article. Below I have some reflections:

- The authors may outline in the article which Candida species are more likely to be associated with mortality in this context.

- Which therapy was most commonly used in patients with CVC, neutropenia, and septic shock?

Author Response

I would like to congratulate the authors on this excellent article. Below I have some reflections:

- The authors may outline in the article which Candida species are more likely to be associated with mortality in this context.

-> As recommended by the reviewer, we did an additional analysis of Candida species in relation to the 60-day mortality. In the propensity matched patients, there was a borderline significant trend of more patients with C. tropicalis in the non-survivor group than in the survivor group (48.3% vs. 27.5%, P=0.060). However, in a multiple logistic regression analysis, there was no significant association between Candida species and the 60-day mortality. We added this information in the results.

“…Regarding the distribution of Candida species, there was a borderline significant trend of more patients with C. tropicalis in the non-survivor group than in the survivor group (48.3% vs. 27.5%, P=0.060)…”

“…Also, specific Candida species was not significantly associated with the 60-day mortality (For C. tropicalis, OR 1.130, 95% CI 0.314 – 4.067, P=0.852, and for C. parapsilosis, OR 0.475, 95% CI 0.091 – 2.491, P=0.379)…”

- Which therapy was most commonly used in patients with CVC, neutropenia, and septic shock?

-> In the unmatched cohort, there were 47 patients with septic shock, 83 patients with CVC, and 8 patients with neutropenia. 1) for 47 patients with septic shock, 38 patients received echinocandins (38/47, 80.9%); 2) for 83 patients with CVC, 60 patients received echinocandins (60/83, 72.3%); 3) for 8 patients with neutropenia, 7 patients received echinocandins (7/8, 87.5%).

In the matched cohort, there were 19 patients with septic shock, 48 patients with CVC, and 6 patients with neutropenia. 1) for 19 patients with septic shock, 10 patients received echinocandins (10/19, 52.6%); 2) for 48 patients with CVC, 25 patients received echinocandins (25/48, 52.1%); 3) for 6 patients with neutropenia, 5 patients received echinocandins (5/6, 83.3%).

Although above information can be calculated from the Table 1, we added the additional explanation regarding septic shock patients in the unmatched cohort in the results.

“…Of note, for 47 patients with septic shock, 38 patients received echinocandins (38/47, 80.9%)…”

Round 2

Reviewer 2 Report

1. The exclusion criteria problem: In that case, the author must delete the redundant exclusion criteria from the manuscript.

2. The recategorization problem: The score-matching analysis has to be repeated with the new research question (60-day survivor VS non-survivor)

Author Response

  1. The exclusion criteria problem: In that case, the author must delete the redundant exclusion criteria from the manuscript.

-> As recommended by the reviewer, previously written exclusion criteria was deleted and we added the revised statement in the manuscript.

“…Only patients who met all of the inclusion criteria were included in the study…”

  1. The recategorization problem: The score-matching analysis has to be repeated with the new research question (60-day survivor VS non-survivor). 

-> Our purpose of a propensity score matching was to control bias/variables between the fluconazole treatment group and echinocandins treatment group. Thus, as a result of a propensity score matching, we came to have a matched cohort (40 patients with fluconazole treatment group and 40 patients with echinocandins treatment group). To calculate 60-day mortality in the matched cohort, we categorized the matched cohort into the 60-day survivor group and 60-day non-survivor group, then performed the comparison analysis and the multivariate analysis. Therefore, we believe that there is no need to repeat a propensity matching for calculation of the 60-day mortality in the already matched cohort. For an example, a propensity matching analysis of the 30-day mortality between ertapenem treatment group and other carbapenem treatment group for treatment of bacteremia showed that the mortality analysis in the matched cohort was performed without repetition of a propensity score matching [Thirty-Day Mortality Rates in Patients with Extended-Spectrum β-Lactamase-Producing Enterobacterales Bacteremia Receiving Ertapenem versus Other Carbapenems. Antimicrob Agents Chemother. 2022 Jul 19;66(7):e0028722.].